# Real-Time Tracking of Time-Varying Cable Frequency Based on a Time-Domain Signal Processing Method

**Zhongqi Shi** [1,2,3] 🔘, **Rumian Zhong** [1,2,3,*] 🔘 **and Nan Jin** [1,2,3]

1   Key Laboratory of Urban Safety Risk Monitoring and Early Warning, Ministry of Emergency Management, Shenzhen 518055, China
2   Shenzhen Technology Institute of Urban Public Safety, Shenzhen 518000, China
3   National Science and Technology Institute of Urban Safety Development, Shenzhen 518000, China
*   Correspondence: zhongrumian@163.com

**Abstract:** In this paper, a time-domain signal processing method is proposed to extract the real-time time-varying cable frequency. The proposed conjugate-pair decomposition (CPD) method uses empirical mode decomposition (EMD) to obtain intrinsic mode functions (IMFs), and then the instantaneous frequency can be extracted by post-processing the conjugate-pair of IMFs. Several numerical simulations and a composite cable-stayed bridge experiment are used to validate the accuracy and capability of the proposed method for tracking time-varying cable frequency. Moreover, the proposed method may be further used to assess cable fatigue damage.

**Keywords:** time-domain signal processing; conjugate-pair decomposition; cable frequency; intrinsic mode functions

## 1. Introduction

Cable structures have been widely used in the field of construction and infrastructure, and it is the critical component bearing the main load of large-span stadiums and cable-stayed bridges. Due to corrosion, fatigue load and other accidental loads in the process of long-term service, cable structures always suffer different degrees of damage [1–3]. As the force and its safety condition are closely related to the vibration frequency of the cable, a real-time tracking method of time-varying cable frequency is required [4,5].

In recent years, a number of frequency identification methods have been discussed in the field of the civil, mechanical and aerospace engineering. The frequency domain method is considered to be the most direct method for identifying the structural modes of structures [6]. The peak pick method is considered the most widely used frequency identification method, but it faces difficulties when handling non-stationary signals [7]. The time-frequency domain method has been widely used in structural real-time frequency identification [8]. STFT can extract frequency components at different time intervals, but it has limited frequency resolution in the fixed window; therefore, STFT is not applicable to the extracted non-stationary signal. The wavelet transform [9] uses the wavelet function to extract the time-domain component simultaneously, which can adjust the window function to narrow the window size. However, the successful extraction of time-dependent frequencies largely depends on the wavelet functions and the discretization of the scale. The Hilbert–Huang transform [2] is a combined method of the Hilbert transform (HT) and EMD. Huang et al. [10] used the EMD, which decomposed the non-stationary or nonlinear signals into intrinsic modular functions (IMF), and the post-processing of each IMF can extract the instantaneous frequencies. However, the Gibbs' effect may cause the extracted frequency to be inaccurate. Hence, new IF tracking methods are necessary [11]. Overall, the above time-frequency analysis method requires time series data covering the minimum frequency signal component for at least three cycles and that the extracted frequencies are the average of all sampled parts.

Based on EMD and least square fitting method, Nayfeh and Pai [12] define instantaneous frequencies and propose a post-processing method to extract parameters of nonlinear structural dynamics. Furthermore, based on the HHT method, Pai et al. [13,14] discussed a modified EMD method, which can extract the instantaneous frequency of structures needing only three points. Recently, Zhong and Pai (2016, 2018) [3,15] proposed a sliding-window tracking-based method to identify the natural frequency of periodic signals.

Based on previous studies, this paper aims to present a time-domain signal processing method to extract the time-varying cable frequency in real-time. In Section 2, the time-domain signal processing method is introduced. In Section 3, several numerical simulations are employed to evaluate the capability of the proposed method for the extraction of time-variant and time-varying frequencies. Section 4 demonstrates the time-varying cable frequency identification method with a composite cable-stayed bridge experiment.

## 2. Time-Domain Signal Processing Method

Conjugate-pair decomposition (CPD) is an indirect time-domain method. Firstly, EMD is used to decompose a nonstationary or nonlinear signal into IMFs which are zero-mean regular or distorted harmonics. As shown in Equation (1):

$$u(t) = \sum_{i=1}^{n} c_i(t) + r_n(t) \tag{1}$$

After a number of iterations, the averaged values of the upper and lower envelopes $m_{1i}$ are computed, and $m_{1k} \approx 0$ [3]; therefore, the first IMF $c_1$ can be extracted as

$$c_1 = u - n_{11} \cdots - n_{1j} \tag{2}$$

Furthermore, the residual $r_1$ can be used to sequentially decompose the results, and $c_i (i = 2, \ldots, m)$ can be extracted as

$$c_i = r_{i-1} - n_{i1} \cdots - n_{ij} \tag{3}$$

$$r_{i-1} = u(t) - c_1 \cdots - c_{i-1} \tag{4}$$

The entire sequential decomposition process will stop as the residual $r_n$ becomes a monotonic function. For example, as shown in Figure 1, $u(t)$ is the displacement, based on the FFT method, and the natural frequencies can be obtained as 7.8 Hz, 9.2 Hz and 12.5 Hz. Moreover, the IMFs of $u(t)$ can be obtained based on EMD, as shown in Figure 1.

Furthermore, the extracted IMF $c(t)$ can be assumed as

$$c(t) = C_0 + e_1 \cos(\omega_1 t) - f_1 \sin(\omega_1 t) = C_0 + a_1 \cos(\omega t + \phi_1) \tag{5}$$

where $C_0$, $e_1$ and $f_1$ are unknown constants, and $f_1 \sin(\omega_1 t)$ is the Hilbert transform of $e_1 \cos(\omega_1 t)$. Then, $t_s$ can be set as the central point of $c(t)$, and the shifted IMF $c(\bar{t})$ can be obtained as follows:

$$\begin{aligned} c(\bar{t}) &= C_0 + C_1 \cos(\omega_1 \bar{t}) - D_1 \sin(\omega_1 \bar{t}) \\ C_1 &= a_1 \cos(\omega_1 t_s + \phi_1), \ D_1 = a_1 \sin(\omega_1 t_s + \phi_1) \\ \bar{t} &= t - t_s \end{aligned} \tag{6}$$

Bases on the least square fitting method (LSF), the three unknown constants $C_0$, $C_1$ and $D_1$ can be computed by minimizing the error function *Error* as:

$$Error = \sum_{i=-(m-1)/2}^{(m-1)/2} [U(t_{n+i}) - c(\bar{t}_i)]^2 \tag{7}$$

where $\bar{t} = \bar{t}_i = i\Delta t$ and $t = t_s + i\Delta t$. Furthermore, the amplitude can be obtained as follows:

$$a_1 = \sqrt{C_1^2 + D_1^2}, \theta_1 = \omega_1 t_n + \phi_1 = \tan^{-1}(D_1/C_1) \tag{8}$$

Then, $t\omega$ can be extracted as

$$\omega = \frac{d\theta}{dt} = \frac{d(\tan^{-1} D_1/C_1)}{dt} \tag{9}$$

To initiate the instantaneous frequency tracking process, the initial value of $\omega$ can be obtained by FFT or HHT.

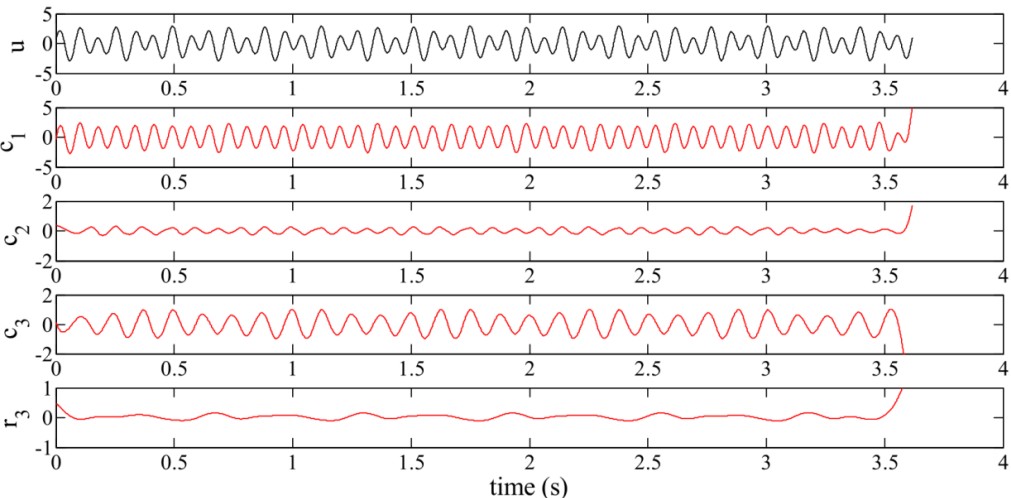

**Figure 1.** Empirical Mode Decomposition.

## 3. Numerical Results and Discussions

### 3.1. Time-Invariant Frequency Extraction

To illustrate the time-domain signal processing method, a two degrees of freedom system is shown as Figure 2. $k$ and $k_0$ denote a linear spring, and the masses are $m_1 = m_2 = 1$. Then, the governing equations can be obtained as

$$\begin{aligned} \ddot{u}_1 + ku_1 + k_0(u_1 - u_2) &= 0 \\ \ddot{u}_2 + ku_2 - k_0(u_1 - u_2) &= 0 \end{aligned} \tag{10}$$

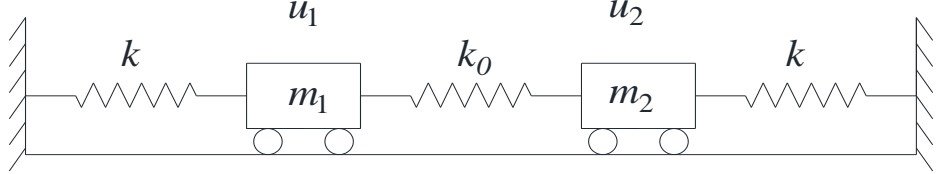

**Figure 2.** Time-invariant frequency extraction system.

Then, $u_1$ can be obtained as

$$u_1 = A \cos \omega_1 t + A \cos \omega_2 t \tag{11}$$

If $\omega_1 = \omega_0 + \xi, \omega_2 = \omega_0 - \xi$, we obtain

$$u_1 = A \cos(\omega_0 + \xi)t + A \cos(\omega_0 - \xi)t = 2A \cos(\xi t) \cos(\omega_0 t) \tag{12}$$

For Equation (12) with $A = 1$, $\omega_0 = 2 \times 2\pi$ Hz, and $\xi = 1 \times 2\pi$ Hz, the frequency response function (FRF) of $u(t)$ can be calculated, as shown in Figure 3b. Moreover, EMD is employed to extracted IMFs $c_1(t)$ and $c_2(t)$, and the initial values of $\omega_1$ and $\omega_2$ are

assumed as 3 Hz and 1 Hz, respectively, based on FFT. Figure 3d,e compares the extracted frequencies based on HHT and CPD. The HHT has severe edge effects, so the CPD method is better extracting the frequency, at least for this case.

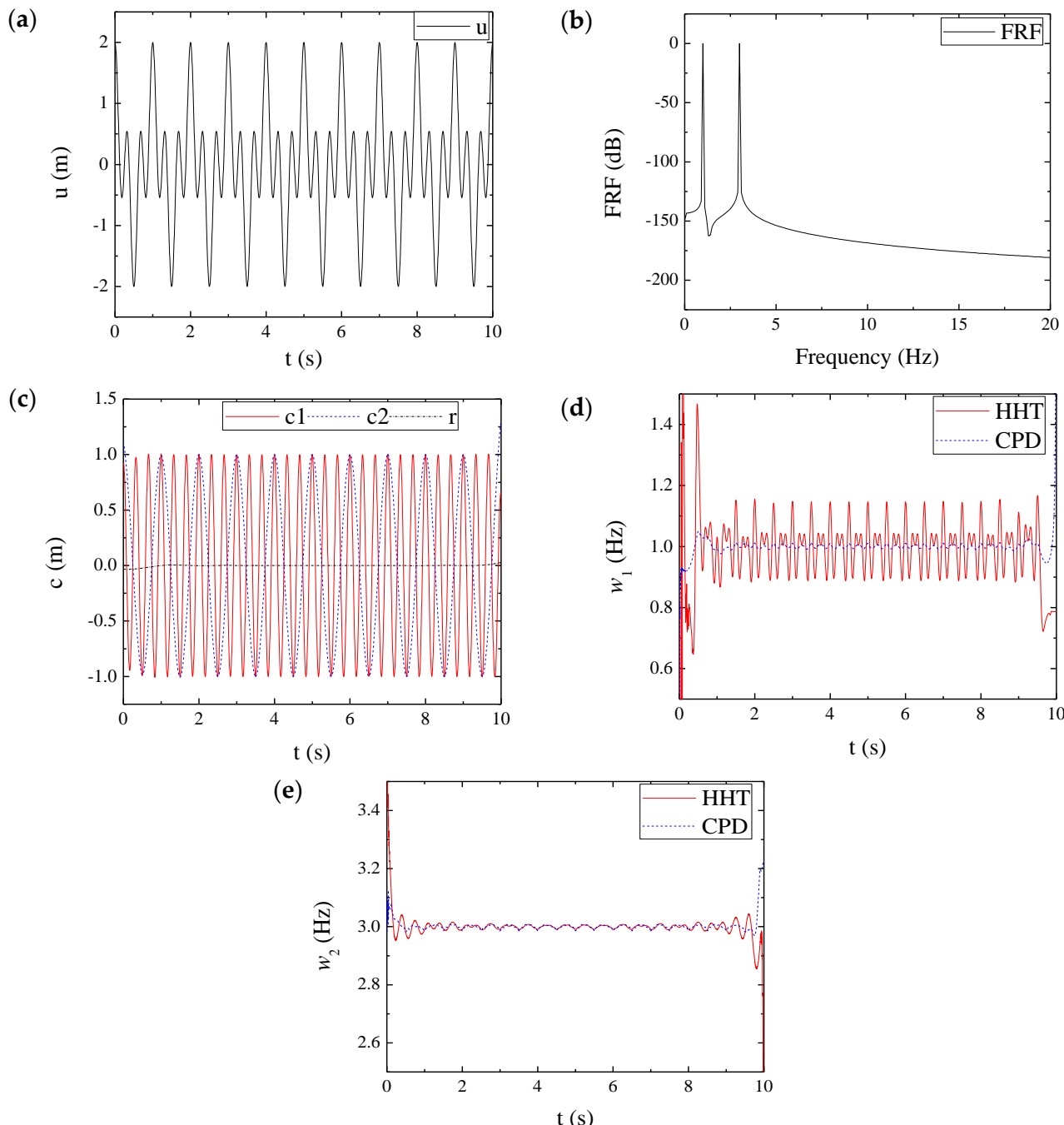

**Figure 3.** Extraction of time-invariant frequency: (**a**) $u(t)$; (**b**) FRF; (**c**) EMD; (**d**) $\omega_1$; (**e**) $\omega_2$.

Furthermore, for Equation (12) with $\xi = 0.0001 \times 2\pi$ Hz $\to 0$ Hz, but $\xi \neq 0$, the amplitude $2A\cos(\xi t)$ changes sign, EMD analysis of the $u_1$ in Equation (11) can only extract one IMF $c_1$, and Figure 4d shows the frequency can be extracted.

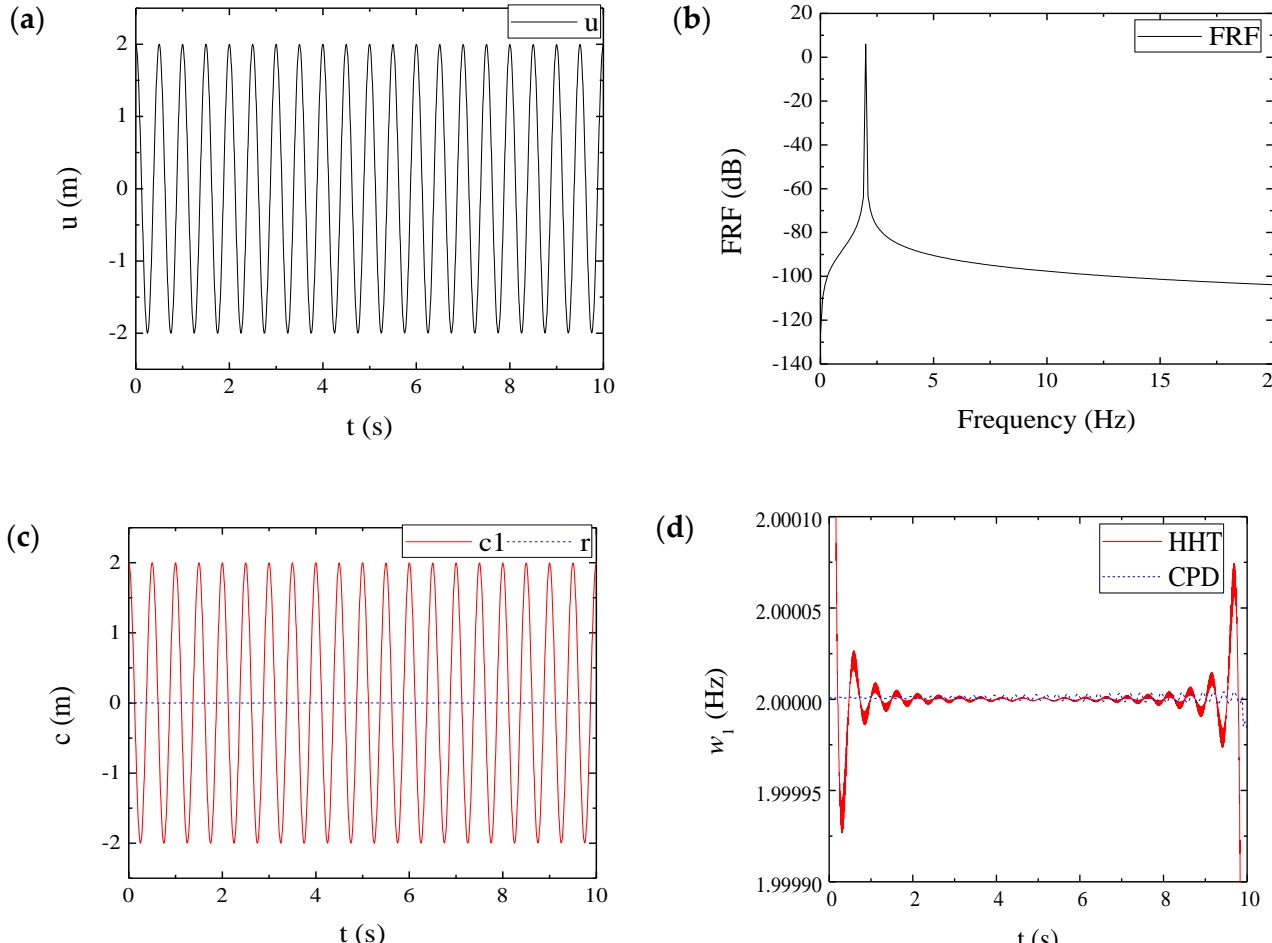

**Figure 4.** Extraction of time-invariant frequency: (**a**) $u(t)$; (**b**) FRF; (**c**) EMD; (**d**) $\omega_1$.

### 3.2. Amplitude Modulation and Frequency Modulation Signals

To demonstrate the instantaneous frequency identification method, we consider an AM-FM signal $u(t)$, as shown in Equation (13), where $C_0 = 1 - e^{-0.1t}$, $\omega_0 = 1$ Hz, $\omega_{AM} = 0.1$ Hz, $\omega_{FM} = 0.2$ Hz, $\omega_i = i$ Hz:

$$u(t) = C_0 + (1 + 0.5\sin 2\pi\omega_{AM}t)\sin(2\pi\omega_0 t + \cos 2\pi\omega_{FM}t) + Noise$$

$$Noise = 0.2 \times \left( \sum_{i=10}^{20} \sin(2\pi\omega_i t + \cos 2\pi\omega_{FM}t) + white\ noise \right)$$

(13)

The FRF of $u(t)$ can be obtained as shown in Figure 5a,b. Figure 5c compared the identified frequency by HHT and CPD. The extracted CPD frequencies are close to the HHT, so they can cooperate and complement each other.

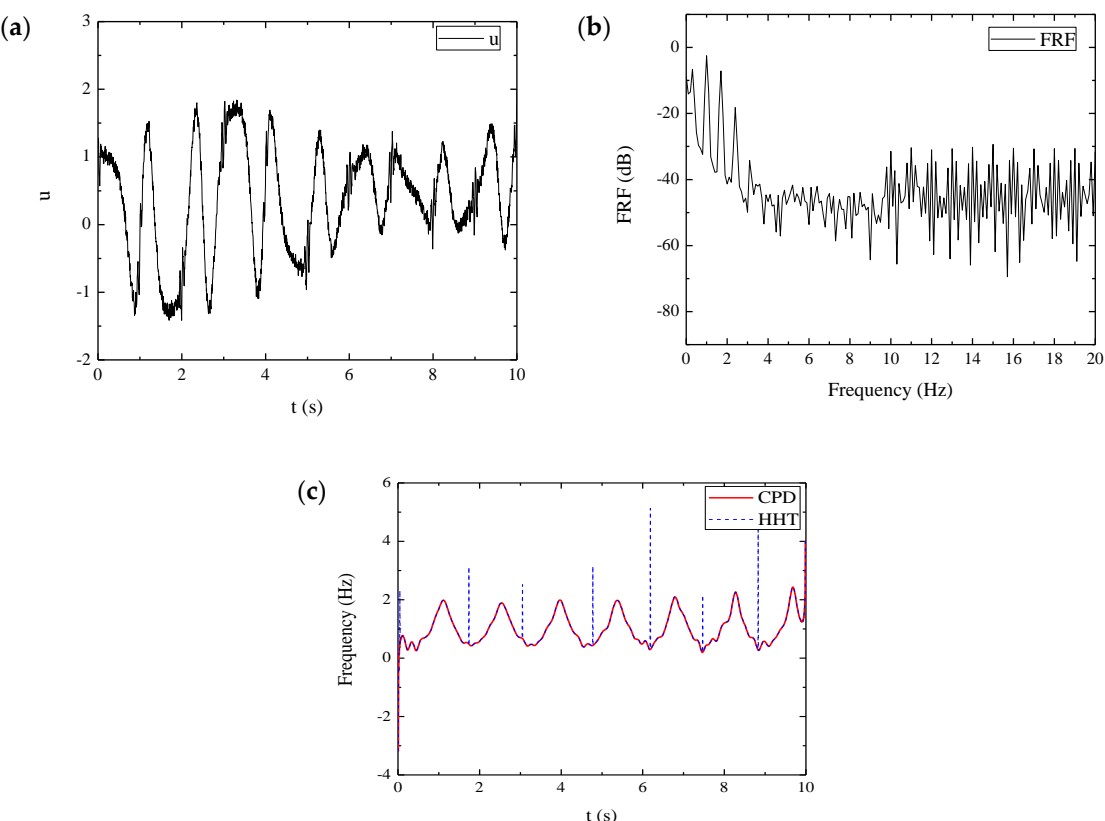

**Figure 5.** AM-FM signal: (**a**) $u(t)$; (**b**) FRF ($u(t)$); (**c**) instantaneous frequency.

## 4. Real-Time Tracking of the Cable Frequency

### 4.1. Benchmark Bridge

The benchmark bridge [15] is shown in Figure 6. In this section, the CPD method is used on a real stay cable (No. 1) based on the structural health monitoring system (Figure 7). Cable No. 1 has a length $l = 217.5$ m, cross-sectional area $A = 0.0105$ m$^2$, elasticity modulus $E = 1.95 \times 10^5$ MPa, and density $\rho = 7800$ kg/m$^3$.

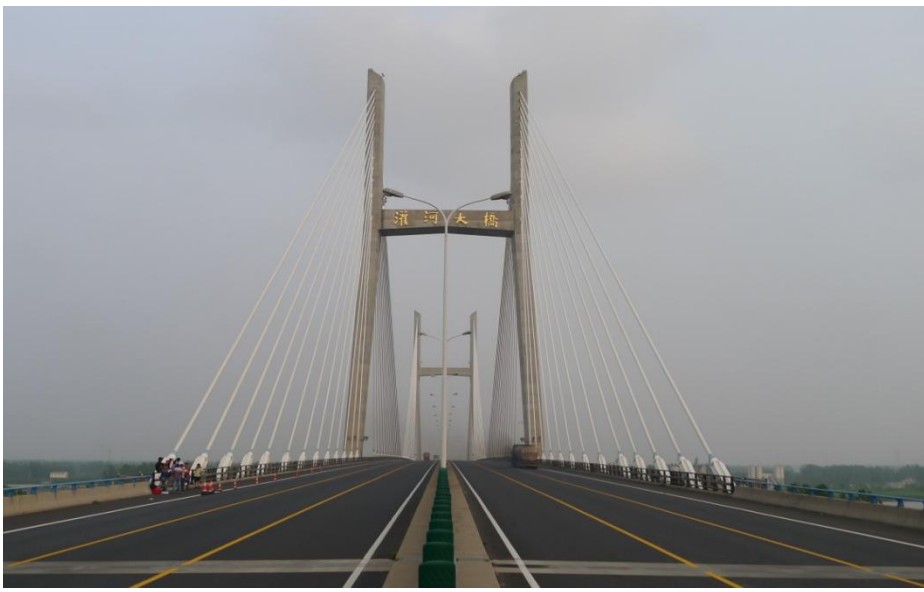

**Figure 6.** Benchmark bridge.

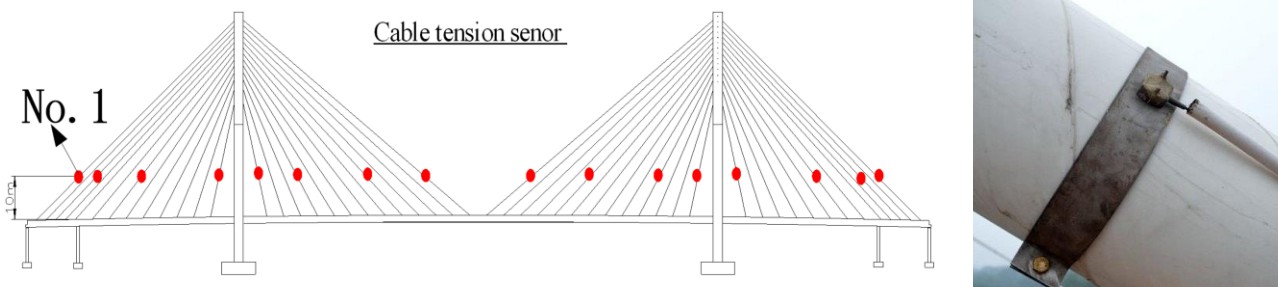

**Figure 7.** Sensors setting of the SHM system.

### 4.2. Time-Domain Signal Processing of a Tensioned Cable

Many numerical simulations have been employed to verify the real-time tracking capability of the CPD method. In this section, the use of the CPD method is examined tracking the frequency of the stay cable under random loading.

Based on the SHM system, the tensioned cable transient vibrations (Figure 8a) were measured using accelerometers (as shown in Table 1). Moreover, the FFT method can be used to estimate the initial value of the first three natural frequencies. As shown in Figure 8b, $f_1 = 0.77$ Hz, $f_2 = 1.55$ Hz, and $f_3 = 2.31$ Hz.

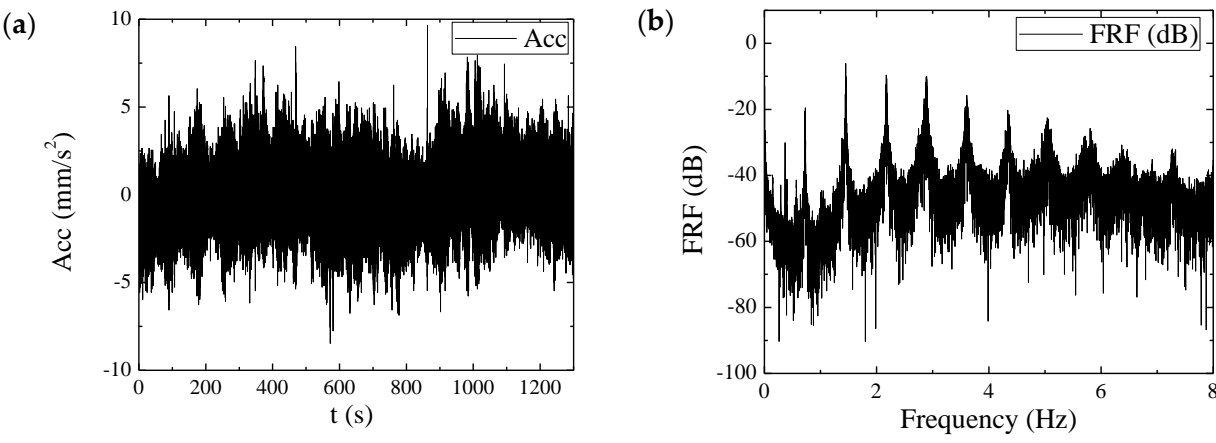

**Figure 8.** The first three natural frequencies: (**a**) Acc, (**b**) FRF.

**Table 1.** Technical parameters of accelerometer.

| Passband (Hz) | Sampling Frequency (Hz) | Resolution Ratio (m/s²) | Loading Resistance (kΩ) | Operating Environment (°C) |
|---|---|---|---|---|
| 0.25, 80 | 50 | $5 \times 10^{-6}$ | 1000 | $-35\,°\mathrm{C}, +70\,°\mathrm{C}$ |

EMD can be performed on the measured acceleration, as shown in Figure 9a. Then, FFT filters are used to eliminate unwanted noise, and the cutoff frequency $f_0 = 2.7$ Hz. Next, the new transient vibration $x(t)$ and FRF can be calculated as Figure 9b,c. Furthermore, the time-varying frequency is extracted as shown in Figure 9h, the extracted frequencies are proportional to the peak of the accelerometer, approximately. When a full truck passes over the bridge, it will cause a larger acceleration of the bridge. On the other hand, the truck can also lead to the increase in cable internal force, and the internal force is proportional to the vibration frequency of the cable. Therefore, the extracted frequency of CPD is consistent with the actual phenomenon.

Because the obtained frequencies are instantaneous values (not averaged values), the relationship between instantaneous frequencies and temperature, wind speed, and vehicle load of the benchmark bridge can be summarized based on the SHM systems and the Weigh In Motion (WIM) system, as shown in Figure 10a–c. In previous studies [3,16], the PP method and SSI method were employed to extract the natural frequencies (3rd order) of the cables, as shown in Figure 10d. Because the temperature, wind speed, and vehicle load changes over time, but the extracted frequency is the average of all sampled parts (t = 30 min), the averaged frequencies are not accurate, and it is difficult to find the relationship between the averaged frequencies and temperature, wind speed, and vehicle load. Therefore, CPD is necessary to track the time-varying cable frequency. Figure 10e shows that the extracted frequency of the cable is proportional to the vehicle load, as shown in Figure 10a. Based on the linear fitting method, the relationship between the instantaneous frequencies and temperature, wind speed, and vehicle load can be obtained, as shown in Figure 10f–h. It is clear that: (1) the extracted frequency is proportional to wind speed and vehicle load; (2) the extracted frequency is inversely proportional to temperature, approximately. As the extracted frequency based on CPD is the instantaneous value, this method can be further applied to study the factors affecting cable structure frequency and damage identification in the future.

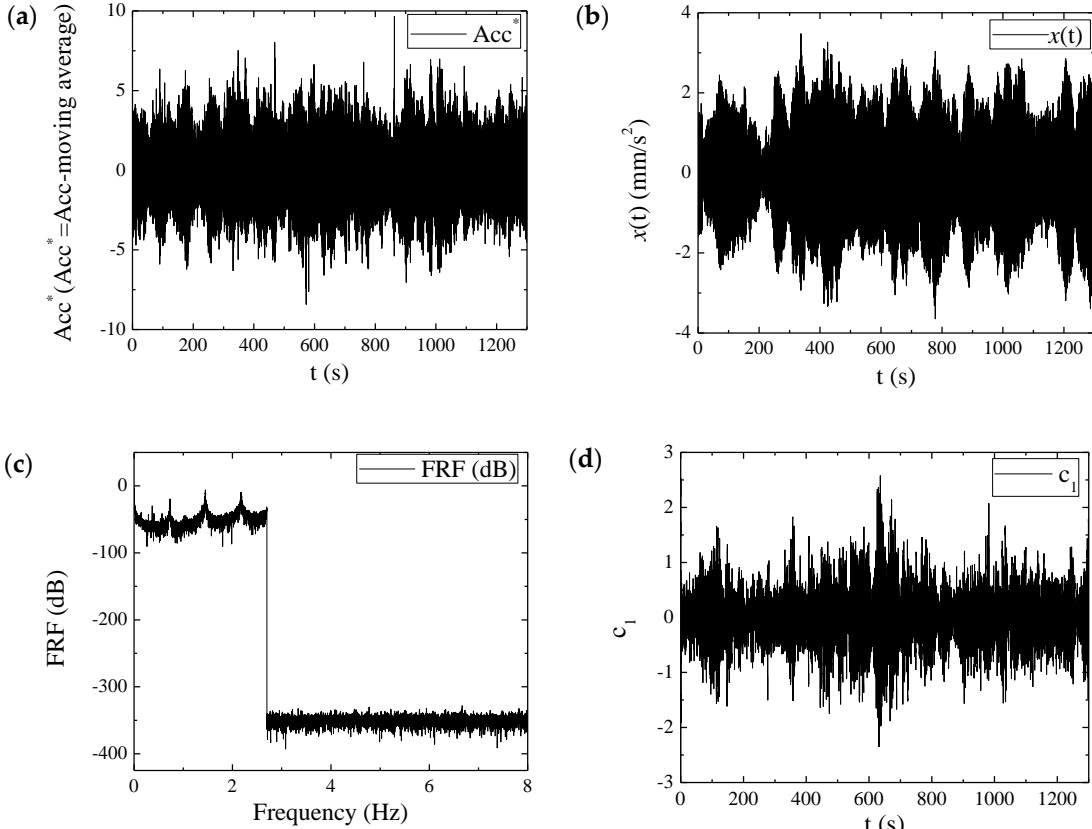

**Figure 9.** *Cont.*

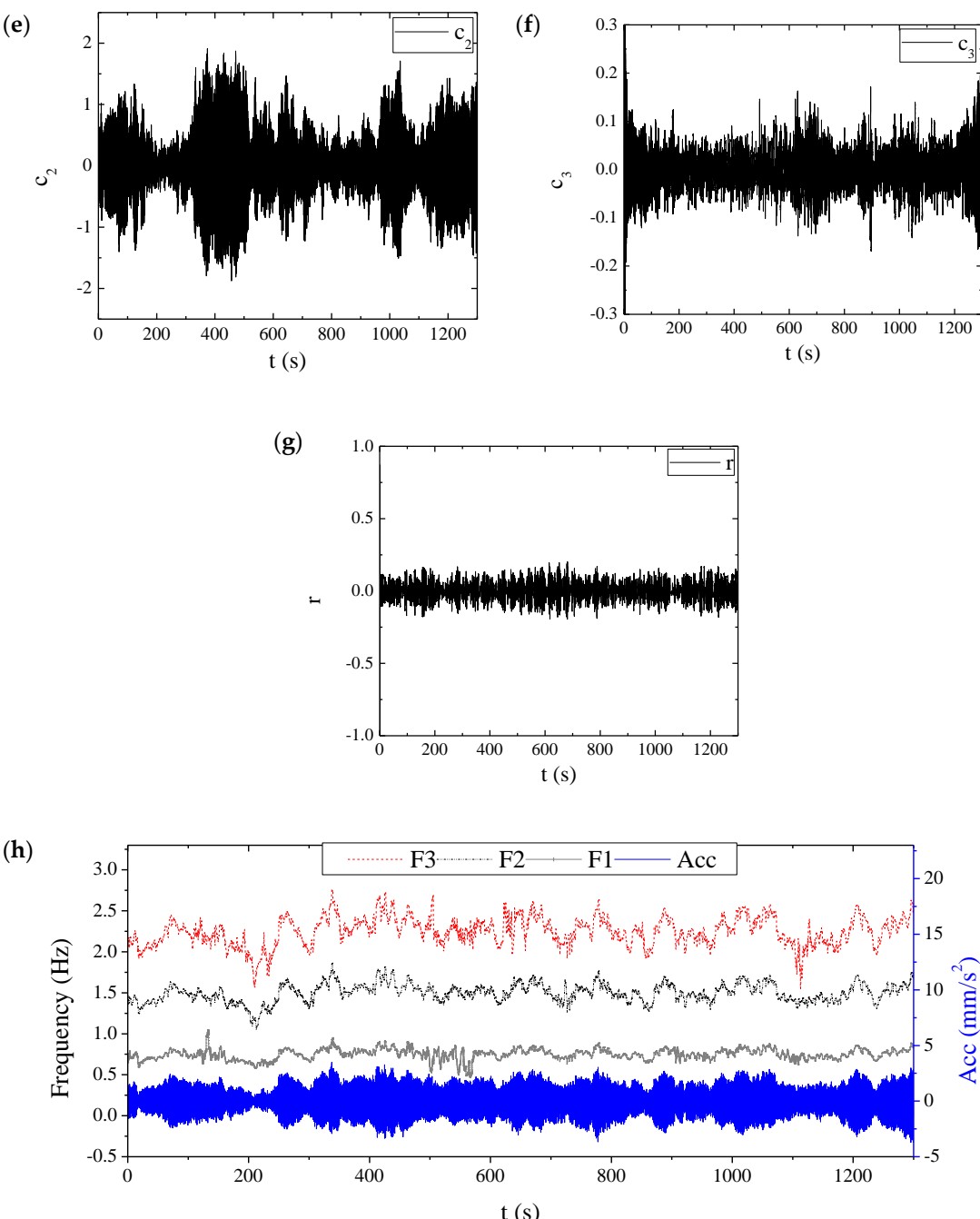

**Figure 9.** Cable frequency: (**a**) $Acc^*$; (**b**) $x(t)$; (**c**) FRF; (**d**) $c_1$; (**e**) $c_2$; (**f**) $c_3$; (**g**) $r_1$; (**h**) frequency.

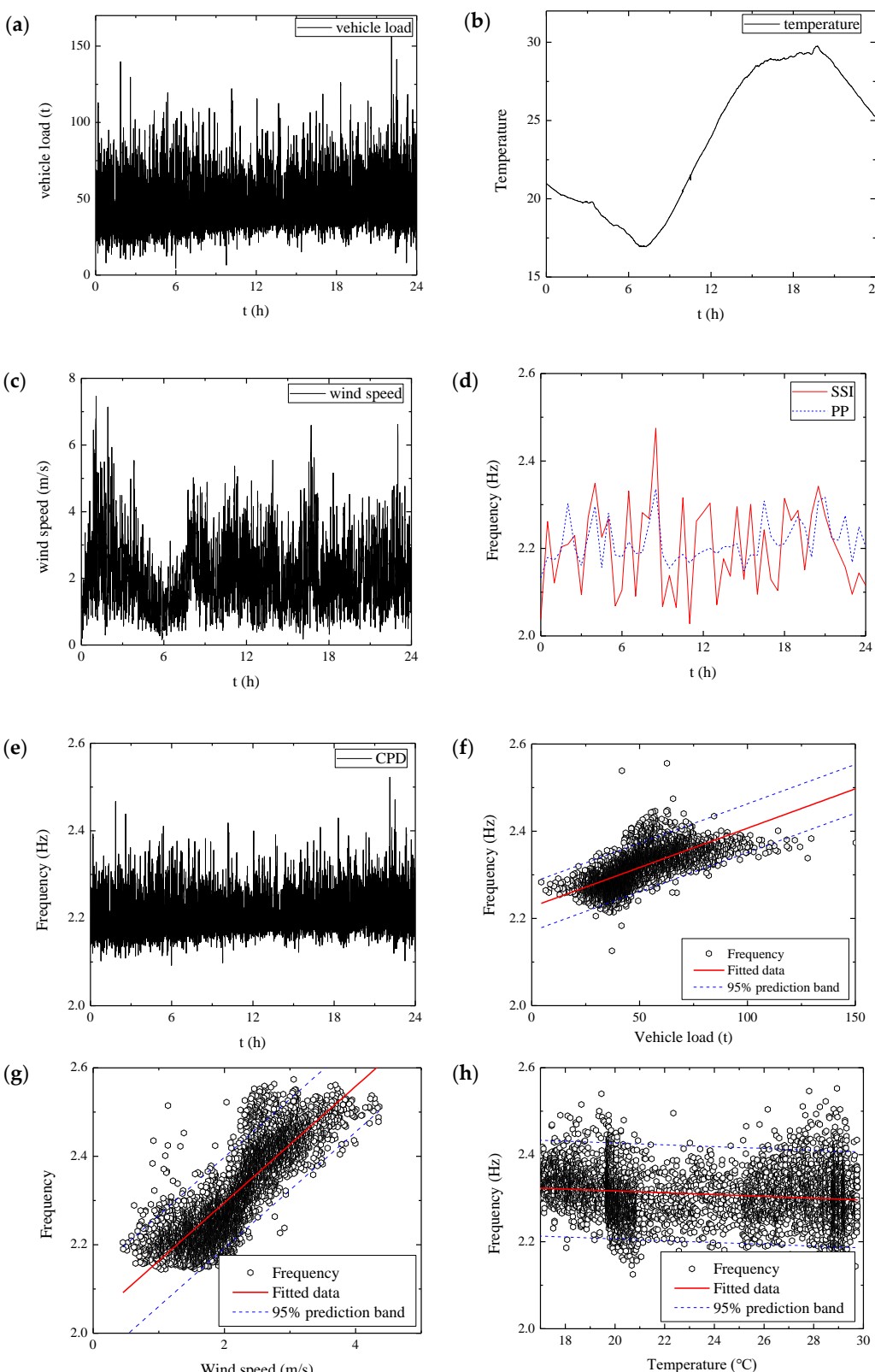

**Figure 10.** Instantaneous frequency identification of stay cable: (**a**) vehicle load; (**b**) temperature; (**c**) wind speed; (**d**) SSI and PP; (**e**) CPD; (**f**) vehicle load; (**g**) wind speed; (**h**) temperature.

## 5. Conclusions

In this paper, a time-domain method is proposed to extract the time-varying cable frequency of bridge structures. Based on several numerical simulations and a cable-stayed bridge experiment, the following conclusions can be drawn.

The CPD method can be used to extract the time-varying cable frequency under random loading, and the extracted frequencies of CPD and HHT can cooperate and complement each other.

The benchmark bridge case validates the accuracy and capability of the CPD method, and the extracted frequencies are proportional to the peak of the accelerometer. Therefore, this method can be further applied to study the factors affecting the cable structure frequency and damage identification in the future.

**Author Contributions:** Conceptualization, Z.S.; Methodology, Z.S., R.Z. and N.J.; Software, R.Z.; Validation, Z.S.; Formal analysis, N.J.; Data curation, R.Z. and N.J.; Writing—original draft, R.Z. All authors have read and agreed to the published version of the manuscript.

**Funding:** The financial support provided by the National Natural Science Foundation of China (NO. 52192664).

**Institutional Review Board Statement:** Not applicable.

**Informed Consent Statement:** Not applicable.

**Data Availability Statement:** Not applicable.

**Conflicts of Interest:** The authors declare no conflict of interest.

## Nomenclature

| | | | |
|---|---|---|---|
| $c_i$ | the ith IMF | EMD | empirical mode decomposition |
| $r_n$ | the residual based on EMD | CPD | conjugate-pair decomposition |
| $\omega_n$ | natural frequency | IMF | intrinsic mode function |
| $k$ | stiffness | HHT | Hilbert–Huang transform |

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
