# Peer review of "Real-Time Tracking of Time-Varying Cable Frequency Based on a Time-Domain Signal Processing Method"

_sustainability, doi:10.3390/su15021700_

Round 1

Reviewer 1 Report

The authors develop a time-domain signal processing methodology for the tracking of the instantaneous frequencies of an arbitrary signal. The presented methodology extends previously published material of the team and it falls quite well into the aims and scope of the journal. The manuscript is, in general, well written and concise, while the case studies support well the theoretical material. I therefore believe that the manuscript should be accepted for publication. In what follows, I summarize a number of individual remarks, which I think that they could help to improve general impression of the manuscript.

1.        A nomenclature/abbreviation table would help a lot in dealing with all the methodological and mathematical notations that are spread all over the text. Moreover, the mathematical notation should be consistent throughout the text.

2.        Line 47-48: I have seen some studies which claims to be successfully tracking IF using HT. The author's claim seems to be a statement which is too strong.

3.        Line 162: capablity Line 162: 'capablity' should be 'capability'

4.        Also, it will be helpful if author can explain how CPD is different from HT. I saw a publication from Dr. Pai where he says that 'signal's conjugate part is obtained from Hilbert transform (HT)'
http://rcada.ncu.edu.tw/2010%20Vol.2_No.1/03.%20CIRCULAR%20INSTANTANEOUS%20FREQUENCY.pdf

5.        After every equation makes sure that any newly introduced quantity is appropriately defined and when the paragraph continues, the word starts from lowercase letter without indentation. Indicatively, both remarks apply to Eq.1, in which many quantities are not defined, while the text after start with a new paragraph (e.g. Where…).

6.        If one intends to use CPD, one has to wait for enough data to do FFT, PP or SSI to get and initial guess and wait till filter reaches steady state?

Reviewer 2 Report

The paper “Real-Time Tracking of Time-Varying Cable Frequency Based on a Time-Domain Signal Processing Method presents a numerical study on a time-domain signal processing method proposed for tracking of the time-varying cable frequency in cable bridges. To verify the efficiency of the proposed method, the measured accelerations of a composite cable-stayed bridge were used. However, this paper is poorly prepared and written. Given the applications and results of the proposed method as well as the technical content of the paper, I think that this paper is more appropriate for a conference paper or technical note. In the reviewer's opinion, this paper is not suitable for publication in Sustainability.

Reviewer 3 Report

Dear Authors,

thank you for this interesting paper.

Please find my comments/suggestions below:

1) Lines 31-35: This sentence is too long, please try to reshape it.

2) Generally, the plots in the paper are too big, please try to make them smaller and organize them in a matrix shape. Currently, it is hard to read the paper since much scrolling is needed.

3) Figure 7 shows the locations of the cable tension sensors on the bridge concerned. On the following Figure 8 you are indicating accelerations. How and where were these accelerations measured? Namely, this data are not originating from the cable tension sensors. What type of accelerometers were used for these measurements and what are their specs (range, accuracy, noise level, resolution etc.)?

4) Lines 170-174: Did you perform some kind of calibration tests for truck driving over the bridge or is this just a general remark? Please indicate this in the text.

5) Did you perform some kind of measurement uncertainity analysis for the frequency determination?

6) Can you please explain the shift around 2,3 Hz in the plot in Figure 9 (c)?

7) Why did you choose to analyze a 1300 s time frame for plots in Figure 9?

8) Lines 189-191: Please note that there are no [46] and [47] references in the paper. Under "natural frequencies of the benchmark bridge" you are considering the bridge superstructure frequencies? If so, please explain how are these relevant for your paper and how where they determined.

Round 2

Reviewer 3 Report

Dear Authors,

thank you for your answers.

However, I have some additional comments.

Regarding my comment no. 3, please explain where were the accelerometers positioned? I don't understand your statement that you have obtained acceleration values from cable tension sensors. How?

Additionally, please address the second part of my comment no. 8 since you obviously skipped it.
